# Comparison Between Worker and Soldier Transcriptomes of Termite *Neotermes binovatus* Reveals Caste Specialization of Host–Flagellate Symbiotic System

**DOI:** 10.3390/insects16030325

**Published:** 2025-03-19

**Authors:** Yu-Hao Huang, Miao Wang, Xiu-Ping Chang, Yun-Ling Ke, Zhi-Qiang Li

**Affiliations:** 1Guangdong Key Laboratory of Animal Conservation and Resource Utilization, Guangdong Public Laboratory of Wild Animal Conservation and Utilization, Institute of Zoology, Guangdong Academy of Sciences, Guangzhou 510260, China; huangyh45@giz.gd.cn (Y.-H.H.); keyl@giz.gd.cn (Y.-L.K.); 2College of Life Science, Shaanxi Normal University, Xi’an 710062, China; 3Shaanxi Key Laboratory for Animal Conservation, Northwest University, Xi’an 710069, China

**Keywords:** termite, *Neotermes binovatus*, transcriptome, caste, flagellate, worker caste, soldier caste, Kalotermitidae, protist, host–symbiont system

## Abstract

Gene expression patterns biased toward workers and soldiers in the host–flagellate symbiotic system have rarely been explored in lower termites. To investigate the caste-specific differences in this system, we sequenced high-depth transcriptomes from the workers and soldiers of the lower termites *Neotermes binovatus*. The functions of caste-biased transcripts align with the defensive role of soldiers and the primary duties of workers, including nest maintenance, preliminary food processing, and communication. Stronger expression correlations between termite and flagellate transcripts were observed in workers compared to soldiers, suggesting a more synergistic relationship in the worker–flagellate symbiotic system for preliminary food processing. This study offers new insights into the functional specialization of the host–flagellate symbiotic system in termite castes.

## 1. Introduction

Termites are a major group of eusocial insects characterized by caste differentiation and polyphenism within the same nest [1,2]. The sterile castes—workers and soldiers—are the most common members of a termite colony. Workers perform all the essential tasks within the nest, including feeding, grooming, caring for the young, foraging, and nest building [1]. Soldiers, which develop from workers through a presoldier stage, are characterized by distinct head capsules, often with powerful mandibles, and their primary role is defending the colony [1,2].

Worker and soldier termites, which arise from the same genotype, exhibit differential gene expression that contributes to their distinct phenotypic development and functional specialization [2,3,4]. Studies on worker-biased and soldier-biased gene expression through transcriptomic analyses in various termite species, including those in Heterotermitidae [5,6,7,8,9,10], Archotermopsidae [11,12], and Termitidae [6], have identified genes involved in development, immunity, digestion, defense, chemosensation, and social interaction that are differentially expressed between castes.

In addition to Termitidae, often referred to as “higher termites”, other termite groups, known as “lower termites”, host various flagellated protists in their hindguts, a characteristic that distinguishes them from most insects, which typically harbor only bacteria, archaea, and fungi in their guts [13,14]. These flagellates are obligate symbionts and are vertically inherited across the termite phylogeny [15]. They play a crucial role in wood digestion in lower termites [14,16,17]. Flagellates are present in most castes of lower termites, including workers and soldiers [6,18,19,20,21]. Studies on the Heterotermitidae species have shown that flagellates are more abundant and exhibit less variation in workers compared to soldiers, as determined by cell counting or meta-transcriptomic analyses [6,18,19,21]. In terms of gene expression, flagellate-derived cellulases in Heterotermitidae are more abundant in workers than in soldiers, which share the same expression patterns of termite-derived cellulases [22,23]. However, the differences in flagellate communities and gene expression in other lower termite species remain largely unexplored.

Kalotermitidae is the second-largest termite family, comprising genera such as *Neotermes*, *Cryptotermes*, and *Kalotermes*, among others [24]. Termites in this family primarily inhabit dry, non-decayed wood and do not require contact with soil or external moisture sources [25]. True workers are absent in Kalotermitidae; instead, false workers perform labor and can differentiate into other castes [25,26]. The genus *Neotermes*, a member of Kalotermitidae, is distributed across tropical and subtropical regions worldwide, where it excavates galleries in sound dead or living wood [24,25]. Most current research on caste specialization in Kalotermitidae focuses on the differential gene expression between workers and reproductive castes in *Cryptotermes* species [26,27]. However, the differences in host–flagellate symbiotic systems between workers and soldiers in Kalotermitidae, particularly in the *Neotermes* species, remain unclear. Transcriptomes for eukaryotes offer a promising approach to exploring the caste-specific gene expression of the host–flagellate symbiotic system of lower termites, such as *Neotermes*.

In this study, we sequenced high-depth transcriptomes of the workers and soldiers of a *Neotermes* species collected in Hainan, China. Using these transcriptomes, we detected differentially expressed transcripts (DETs), identified the transcripts from the termite and various flagellate orders, and constructed co-expression networks of host and flagellate transcripts in order to investigate specific pattens of the host–flagellate symbiotic system of worker and soldier *Neotermes binovatus*.

## 2. Materials and Methods

### 2.1. RNA Extraction and Transcriptome Sequencing

Workers and soldiers of wild *N. binovatus* were collected from a single colony in Wuzhishan, Hainan, China (18°54′13″ N, 109°40′9″ E). The species was identified through a morphological examination of the soldiers, and three soldier specimens were retained for future morphological verification. Ten workers and ten soldiers were selected, with each individual serving as a separate sample for transcriptome sequencing. The samples were immediately frozen in liquid nitrogen before RNA extraction. The entire body of each frozen individual was homogenized for RNA extraction. Total RNA was extracted from each individual using a TRIzol reagent (CWBIO, Beijing, China). RNA quality and quantity were assessed using a Nanodrop 1000 spectrophotometer (Thermo Fisher Scientific, Wilmington, DE, USA) and an Agilent 2100 bioanalyzer (Agilent Technologies, Santa Clara, CA, USA). All RNA samples (ten workers and ten soldiers) with a 260/280 ratio between 1.8 and 2.0, a 260/230 ratio between 2.0 and 2.5, and an RNA integrity number (RIN) greater than 8.0 were used for library construction and sequencing.

mRNA was isolated using Oligo(dT)-attached magnetic beads and subsequently fragmented randomly in fragmentation buffer. First-strand complementary DNA (cDNA) was synthesized using the fragmented mRNA as a template and random hexamers as primers, followed by second-strand synthesis with the addition of PCR buffer, dNTPs, RNase H, and DNA polymerase I. The cDNA was then purified using AMPure XP beads (Beckman Coulter, Brea, CA, USA). The double-stranded cDNA underwent end repair, with the addition of an adenine tail at the ends and the ligation of adapters. The cDNA library was constructed through the PCR amplification of the cDNA fragments. Sequencing was performed on the Illumina HiSeq 2500 platform (Illumina, San Diego, CA, USA), with a 12 Gb sequencing depth for each sample, generating 2 × 150 bp reads.

### 2.2. Transcriptome Assembly, Protein Prediction, and Functional Annotation

Raw reads were first processed to remove adaptors and low-quality sequences using Trimmomatic v0.36 [28] with the default settings. The transcriptomes of the *N. binovatus* individuals were then assembled de novo using Trinity v2.15.2 [29]. Assembly completeness was evaluated with BUSCO v5.7.1 [30], using the Insecta ortholog set from OrthoDB v10 [31]. To verify our morphological taxonomy identification, we used MitoFinder v1.4.1 [32], along with termite mitochondrial genomes from the National Center for Biotechnology Information (NCBI) nucleotide sequence (NT) database as references. We searched for mitochondrial sequences, predicted the Cytochrome Oxidase 2 (COX2) coding sequences, and compared them against the NT database using BLASTN v2.16.0+ [33]. The COX2 sequences confirmed that the termites belong to *Neotermes*. Species identification could not be confirmed, as no *N. binovatus* sequence was available in the NCBI database.

Nuclear proteins were predicted from the transcript sequences using TransDecoder v5.7.1 (https://github.com/TransDecoder/TransDecoder, accessed on 26 August 2024) and annotated via a modified “annotate” pipeline of FunAnnotate v1.8.17 (https://github.com/nextgenusfs/funannotate, accessed on 26 August 2024). Specifically, Gene Ontology (GO) [34] annotations were obtained by combining results from InterProScan v5.69 (InterPro v101.0 database) [35,36] and Eggnog-mapper v2.1.12 (EggNog v5.0.2 database) [37,38].

### 2.3. Abundance Quantification and Differential Expression Analysis

Using the assembly as a reference, the transcript abundance in each sample was quantified with RSEM v1.3.3 [39], utilizing Bowtie2 v2.5.4 [40] as the aligner. Transcripts with low expression levels were filtered out to minimize false positives in the subsequent abundance analyses. This filtering was performed using a method implemented in the R package edgeR v3.40.2 [41]. In detail, transcripts with more than ten samples showing counts per million (CPM) greater than 1.0 were retained, while others were excluded.

DETs were detected using the R package DESeq2 v1.38.3 [42], with the criteria set at a fold change >2 or <0.5 and an adjusted *p*-value (Q-value) < 0.05. The statistical significance of GO enrichment was assessed using hypergeometric distribution testing in the R package clusterProfiler v4.6.2 [43]. The *p*-values were adjusted for multiple testing using the Benjamini–Hochberg procedure with a cutoff of 0.05. GO-Figure! v1.0.2 [44] was employed to cluster and visualize the significantly enriched GO terms.

Additionally, Least Absolute Shrinkage and Selection Operator (LASSO) regression was used to identify the most important and stable biomarker transcripts distinguishing workers and soldiers of *N. binovatus* based on the Trimmed Mean of M-values (TMM) transcript expression. The TMM normalization of filtered expression counts was carried out using the edgeR package, while the LASSO regression analysis was performed using the R package glmnet v4.1-8 [45].

### 2.4. Sequence Source Identification

After excluding low-expressed transcripts as mentioned above, the remaining transcripts were searched against the NT database using BLASTN v2.16.0+ [33] with an E-value cutoff of 1 × 10^−5^. Transcripts were identified as termite or flagellate sequences based on their best hits. Only those with the best matches to Blattodea (Taxonomy ID: 85823) or Metamonada (Taxonomy ID: 2611341) were identified as termite or flagellate transcripts, respectively. These transcripts should also show no hits against sequences from other organisms in the regions not overlapping with the best hits. Additionally, for flagellate transcripts, we examined the top five hits against the Metamonada sequences with a percent identity greater than 80%. If all the five hits for a transcript were from the same order, the transcript was classified into that order. If the top five hits had a percent identity greater than 90% and were from a single genus, the transcript source was identified as that genus.

### 2.5. Co-Expression Network Construction

We constructed the co-expression network based on the TMM matrix using the R package WGCNA v1.73 [46]. The TMM expression data of workers and soldiers, as described above, were used to perform network construction and consensus module detection following the standard procedure and parameters for consensus WGCNA analysis. The soft-thresholding power was automatically selected by WGCNA and was ultimately set to 12.

## 3. Results

### 3.1. General Features of Transcriptome of N. binovatus

For each worker and soldier sample, clean data ranging from 11.45 to 14.17 Gb were generated using the Illumina platform. The transcriptome of *N. binovatus* was assembled by Trinity [29], resulting in 1,042,913 contigs with an N50 of 915 bp. Of these, 205,108 contigs were annotated in InterPro [35], 195,426 in EggNog [38], and 177,387 in GO [34] database. The Benchmarking Universal Single-Copy Orthologs (BUSCO) pipeline [30] detected 99.0% complete genes, 0.6% fragmented genes, and 0.4% missing genes, indicating exceptionally high assembly integrity.

### 3.2. Differential Expression of Termite Host Transcripts Between Workers and Soldier

After excluding transcripts with low expression, the differential expression analysis of 30,605 transcripts identified 599 worker-biased and 958 soldier-biased transcripts. Among the 19,842 transcripts with putative termite sources, 447 were worker-biased and 566 were soldier-biased. In order to explore the overall functional distribution of caste-biased termite transcripts, we performed GO enrichment analysis, using worker-biased and soldier-biased transcripts as foregrounds and all termite transcripts as the background. GO enrichment analysis revealed that worker-biased termite transcripts were primarily associated with cuticle development, synaptic target inhibition, pheromone biosynthesis, and metabolism of chitin, fatty acid, xenobiotics, and UDP-glucose (Figure 1A). In contrast, soldier-biased termite transcripts were mainly involved in muscle development and kinesis, complex motion, dopamine catabolism, protein modification, and calcium ion transport (Figure 1B).

Using LASSO regression, we identified 12 stable biomarkers distinguishing worker and soldier samples (Table 1). Of these, 10 were detected in the differential expression analysis, including 8 putative termite transcripts as follows: resistance to inhibitors of cholinesterase protein 3 (RIC3), T-box transcription factor TBX20, CAP-Gly domain-containing linker protein 2 (CLIP2), periaxin, regulator of microtubule dynamics protein 1 (RMDN1), choline transporter-like protein 2 (CTL2), sorting nexin-29 (SNX29), and pancreatic triacylglycerol lipase (PTL) mRNAs.

### 3.3. Differences in Flagellate Transcript Abundance Between Workers and Soldiers

A search against the NCBI NT database identified 433 flagellate transcripts, including 94 from Cristamonadida (10 18S rRNAs, 63 actin mRNAs, 16 elongation factor-1 alpha (EF1α) mRNAs, and 5 glyceraldehyde-3-phosphate dehydrogenase (GAPDH) mRNAs), 8 from Oxymonadida (7 18S rRNAs and 1 alpha tubulin mRNA), 10 from Trichomonadida (2 18S rRNAs, 4 actin mRNAs, 1 EF1α mRNA, and 3 other genes), and 12 from Tritrichomonadida (11 actin mRNAs and 1 alpha tubulin mRNA) (Figure 2, Table 2). An additional 309 putative flagellate transcripts could not be classified into a specific order. These included 5 18S rRNAs, 76 actin mRNAs, 48 alpha tubulin mRNAs, 56 beta tubulin mRNAs, 10 EF1α mRNAs, 16 GAPDH mRNAs, and 98 other genes (Table 2).

At the genus level, only 10 18S rRNAs were identified, including 4 from *Devescovina* (Cristamonadida), 5 from *Oxymonas* (Oxymonadida), and 1 from *Blattamonas* (Oxymonadida) (Table 3). No significant differences or caste-specific existence in transcript abundance were observed between the worker and soldier *N. binovatus* (Table 3).

Flagellate transcripts, particularly those from Cristamonadida and Tritrichomonadida, were generally more abundant in workers of *N. binovatus* (Figure 2). However, the abundance of most flagellate transcripts did not significantly differ between workers and soldiers. Only eight flagellate transcripts (two actin mRNAs from Tritrichomonadida, and three actin mRNAs, one alpha tubulin mRNA, one beta tubulin mRNA, and one EF1α mRNA from unclassified order) were significantly more abundant in workers. In contrast, two flagellate transcripts (23S rRNA and beta tubulin mRNA from unclassified order) were biased toward soldiers.

### 3.4. Co-Expression Patterns of Host–Flagellate Transcripts in Workers and Soldiers

A Consensus Weighted Gene Co-Expression Network Analysis (WGCNA) of the transcriptomic expression matrices of the worker and soldier *N. binovatus* revealed the co-expression correlations between putative flagellate and other transcripts in the two castes (Figure 3). In worker *N. binovatus*, 12,742 pairs of putative flagellate and other transcripts exhibited strong correlations (WGCNA edge weight ≥ 0.5), involving 20 transcripts putatively from Cristamonadida, 2 from Trichomonadida, 7 from Tritrichomonadida, 54 from unclassified flagellates, 144 from the termite, 1053 from unclear sources, and none from Oxymonadida (Figure 3A). This was much more than the number of correlated pairs in the soldier caste (Figure 3B). The strongly correlated termite transcripts were predominantly associated with biological processes such as appendage morphogenesis, sex development, cell development, cell interaction, locomotory behavior, organic hydroxy compound metabolism, responses to metal ion and xenobiotic stimulus, and the regulation of immunity, protein secretion, and body fluid levels (Figure 3C). But no terms were significant in the enrichment analysis. In the soldier *N. binovatus*, only 38 strongly correlated pairs (WGCNA edge weight ≥ 0.5) of putative flagellate and other transcripts were identified, involving 3 transcripts putatively from Cristamonadida, 1 from unclassified flagellate, and 17 from unclear sources (Figure 3B). Notably, a strong correlation was observed between an actin mRNA from Cristamonadida and the transcripts of one glycosyl hydrolase (GH) 45, two GH10, and one profilin mRNAs from unclear sources. Additionally, another actin mRNA from Cristamonadida was strongly correlated with the same GH45 and GH10 mRNAs, one lectin domain containing mRNA, and one unknown protein mRNA from unclear sources. Moreover, a malic enzyme mRNA from an unclassified flagellate exhibited strong correlation with the same GH45 and GH10 mRNAs, one GH7 mRNA, one carbohydrate esterase (CE) 4 mRNA, and one unknown protein mRNA from unclear sources. All the above strong correlations were also detected in the worker termites (Figure 3D).

## 4. Discussion

### 4.1. Caste-Biased Transcript Expression Patterns of the Host Worker and Soldier Termites

In this study, we sequenced and compared the worker and soldier transcriptomes of the lower termite *N. binovatus*. The number of soldier-biased transcripts was at least 25% greater than that of the worker-biased transcripts, which appears to align with the higher proportion of soldier-biased genes observed in other termite families, such as Heterotermitidae and Termitidae [6,9]. However, it is important to note that our transcript-level results do not directly correspond to the number of true genes, as multiple differentially expressed transcripts may originate from the same gene. Further validation is required to confirm whether the number of soldier-biased genes is indeed higher.

The worker-biased transcripts of *N. binovatus* were predominantly associated with processes related to cuticle development, nervous system regulation (synaptic target inhibition), pheromone biosynthesis, and metabolism (e.g., chitin, fatty acids, xenobiotics, and UDP-glucose). These findings reflect the roles of workers in nest maintenance, preliminary food processing, and communication [1]. In *Reticulitermes aculabialis* (Heterotermitidae), workers upregulate the endocuticular protein genes to increase the thickness of the endocuticle layers, possibly to enhance their resistance to environmental stress [7]. Similar gene expression patterns in worker *N. binovatus* suggest a possible phenotype of thicker endocuticle layers similar to that observed in the Heterotermitidae species, which requires further measurement. Furthermore, a worker-biased biomarker transcript, CTL2, identified through LASSO regression, interacts with chitin synthase and is involved in insect cuticle development [47]. This transcript may facilitate the development of the endocuticle layers of the workers. Other worker-biased biomarker transcripts are potentially linked to nervous system regulation (periaxin) [48], cognitive ability regulation (RMDN1) [49], and myoblast differentiation (SNX29) [50].

For soldier *N. binovatus*, the transcripts associated with muscle development and kinesis, protein modification, and dopamine catabolism were upregulated. The upregulation of muscle-related genes in soldiers has been previously reported in the Heterotermitidae species [9,22], probably linked to the use of muscular force in defense, particularly through the enlarged mandibles [2]. Transcripts related to protein modification may play a role in caste determination and differentiation in eusocial insects, and they are highly expressed in soldier *Reticulitermes speratus* (Heterotermitidae) [51], which is consistent with our findings. Dopamine, which modulates aggressive behavior in *Drosophila* [52], is also upregulated in soldier termites, supporting its role in defense. Additionally, one of the soldier-biased biomarker transcripts identified by LASSO regression, RIC3, enhances the reception of 5-hydroxytryptamine, potentially increasing aggression [53,54]. Other soldier-biased biomarker transcripts may be involved in leg development (TBX20) [55], nervous system development (CLIP2) [56], and lipid metabolism (PTL) [57].

### 4.2. Flagellate Community Composition in Worker and Soldier Termites

Lower termites, including the *Neotermes* species, harbor flagellates from several orders within the phylum Parabasalia and the order Oxymonodida in Preaxostyla [14]. In our high-depth transcriptomes of *N. binovatus*, we identified flagellate transcripts from multiple orders (i.e., Cristamonadida, Trichomonadida, Tritrichomonadida) in Parabasalia and Oxymonadida. A total of ten transcripts were identified from the genera *Devescovina*, *Oxymonas*, and *Blattamonas*. Cristamonadida and Oxymonadida are the predominant orders of flagellates in the *Neotermes* species, with *Devescovina* and *Oxymonas* recognized as two common genera within *Neotermes* flagellates [14,58]. Tritrichomonadida has also been observed in *Neotermes cubanus*, albeit at low abundance [58], while Trichomonadida has been occasionally observed in Heterotermitidae termites [21]. *Blattamonas* is primarily found in non-termite cockroaches [59]. The appreciable abundance of these flagellate transcripts in both worker and soldier individuals suggests the obligate or occasional presence of these flagellates or their close relatives in the entire *N. binovatus* colony.

Most of the putative flagellate transcripts identified in this study are phylogenetic markers (e.g., 18S, actin, alpha and beta tubulin, EF1α, GAPDH), which are commonly used as reference genes in eukaryotic studies due to their relatively stable expression [60,61,62]. These markers serve as proxies for flagellate abundance; however, their expression levels may not fully correlate with the actual cell counts due to potential variations in transcript stability. The limited identification of most flagellate transcripts suggests that the current sequence database lacks sufficient representation of these organisms. Furthermore, we identified only a few 18S rRNAs from the Oxymonadida, and almost no other Oxymonadida markers were detected, indicating insufficient information for identifying this group through most markers. Similar limitations were observed for alpha and beta tubulin across all orders, as well as for EF1α and GAPDH across all orders except Cristamonadida. At the genus level, only a few 18S rRNAs from three genera were identified. This lack of known flagellate sequences hinders the comprehensive exploration of symbiotic flagellates through termite transcriptomes.

The abundance of most flagellate transcripts tends to be higher in worker *N. binovatus*, although the differences among most of these transcripts are not statistically significant. This trend may partly support the previous findings indicating a higher number of flagellates in the worker termites compared to the soldier termites [6,18,19,21]. This is probably because workers are responsible for preliminary food processing and subsequently feed soldiers with processed food and flagellates through proctodeal trophallaxis [1,15]. However, unlike direct cell counting, which provides strong and reliable results, sequence abundance may be less effective in detecting significant differences in flagellate numbers between workers and soldiers, especially with limited replication. Both our results and qPCR analysis from a previous study [20] show a higher trend of flagellate sequence abundance in worker termites compared to soldiers, though the difference is not statistically significant. This suggests that the two methods may differ in sensitivity. Therefore, the hypothesis that worker *N. binovatus* has a higher number of flagellates should be further validated through cell counting in future studies.

### 4.3. Host–Flagellate Co-Expression Correlation in Worker and Soldier Termites

We analyzed the co-expression patterns of termite and flagellate transcripts in the worker and soldier *N. binovatus* termites and found a more complex co-expression correlation structure in the workers, particularly between the putative termite and flagellate transcripts, compared to the soldiers. This suggests a weaker interaction between the soldier termites and the flagellates. In fact, species-specific symbiotic flagellates in soldier *Coptotermes formosanus* (Heterotermitidae) exhibited greater variability in abundance than those in the workers, and some flagellate species were sometimes absent in soldiers altogether [21]. Integrating these community dynamics with our co-expression correlation findings, it appears that worker termites closely cooperate with their flagellates to preliminarily process food, while soldier termites mainly rely on food processing by workers rather than on flagellates. The flagellates present in soldiers likely play a supportive or incidental role in food digestion.

The putative termite transcripts that correlate with the flagellate transcripts in the workers represent the potential candidate genes involved in host–flagellate interaction. Flagellates can be influenced by termite hormones and die prior to the molt of their host, requiring replenishment through proctodeal trophallaxis [15]. This may explain the host–flagellate correlation observed in processes related to morphogenesis and development. Termite transcripts involved in immune regulation could potentially act on the flagellates or their symbiotic bacteria [63], thereby regulating the flagellate community. Additionally, transcripts related to other functions may contribute to host–flagellate cell surface interaction (cell interaction, locomotory behavior), as well as to changes in the physiological environment of the gut or metabolic interaction (organic hydroxy compound metabolism, responses to metal ion and xenobiotic stimulus, and regulation of protein secretion and body fluid levels). The actual roles of these termite transcripts should be experimentally validated to further explore the potential mechanisms of host–flagellate regulation.

We also observed that the expression of numerous transcripts with unclear sources was strongly correlated with the putative flagellate transcripts, including GH7, GH10, GH45, and CE4 mRNAs, in both the workers and soldiers. These GH and CE transcripts are involved in lignocellulose digestion in termites and are highly expressed in their symbiotic flagellates [16,17], yet they are absent from termite genomes [5,64]. This suggests that these transcripts likely originate from flagellates and could not be identified through sequence similarity searches against current databases. The large number of transcripts with unclear sources may represent such unidentified flagellate transcripts, which act as “dark matter” within lower termite transcriptomes. These transcripts constitute a potential gene pool worthy to be explored in further studies, particularly through the use of an improved database containing comprehensive genomic and transcriptomic assemblies of termite flagellates, such as the draft genome of the oxymonad *Streblomastix strix* from the termite *Zootermopsis angusticollis* [65].

## 5. Conclusions

This study investigated the differential abundance of termite and flagellate transcripts, as well as the co-expression patterns of host–flagellate transcript pairs, between the worker and soldier castes of the lower termite *N. binovatus*. The functions of the worker-biased transcripts reflected the roles of workers in nest maintenance, preliminary food processing, and communication. The soldier-biased transcripts indicated functional specialization for their defensive role. The abundance of most flagellate transcripts tended to be higher in the workers, and a larger number of strong co-expression correlations between the termite and flagellate transcripts were detected in the workers. These findings suggest a potential synergy in preliminary food processing within worker holobionts, which may support the nutritional needs of soldiers without a strong dependence on flagellates in themselves. Overall, these results highlight the potential specialization patterns in the host–flagellate symbiotic system between the worker and soldier castes of lower termites, offering new insights into the caste division of eusocial insects.

However, it should be noted that our findings and proposed explanations are entirely based on transcript-level analyses from a single colony of the species *N. binovatus*. Some of the caste-biased patterns identified may be specific to the genotype of this colony or species and may not apply to other termite colonies or species. Additionally, false positive results could arise from the transcriptomic analyses, particularly in the absence of a reference genome for this species. Therefore, the candidate caste-biased termite and flagellate transcripts, as well as the potential correlations or interactions between host and flagellate transcripts, should be further validated through experiments such as qPCR, RNA interference, flagellate cell counting, and metabolite quantification. Furthermore, the limited availability of termite flagellate sequence resources hindered a more thorough exploration of the flagellate transcripts, underscoring the need for the further sequencing of termite flagellate genomes or transcriptomes in future studies.

## Figures and Tables

**Figure 1 insects-16-00325-f001:**
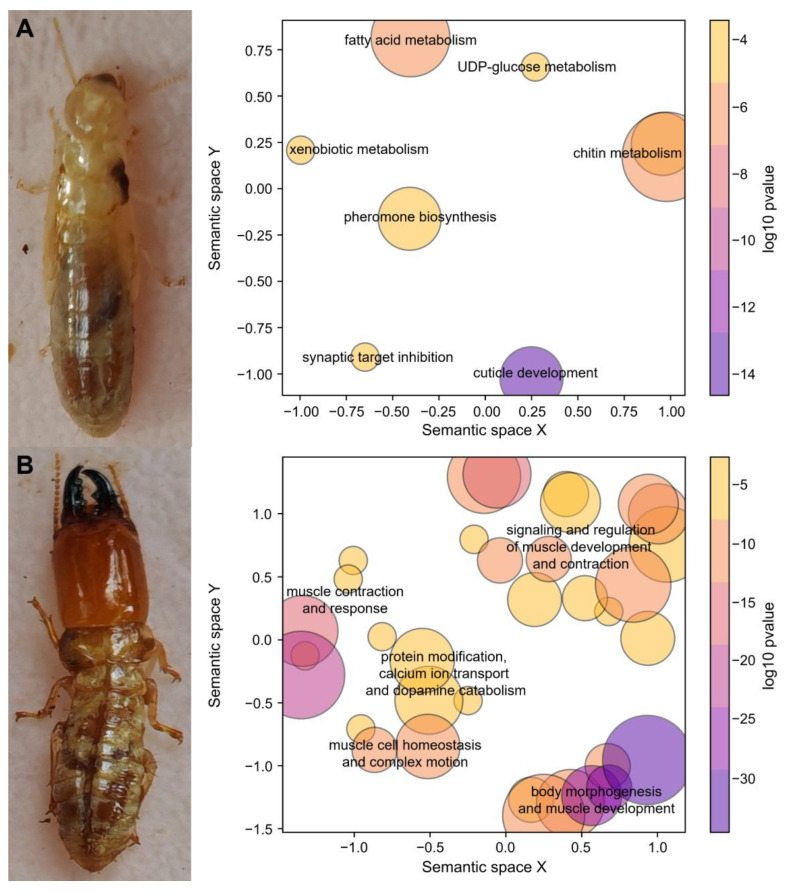
Semantic similarity scatterplots of significantly enriched Gene Ontology (GO) terms in the Biological Process (BP) category, highlighting functions of (**A**) worker-biased termite transcripts and (**B**) soldier-biased termite transcripts. Circles represent clusters of GO terms, with similar clusters positioned closely together. The function descriptions summarize the roles of these clusters. The size of the circles corresponds to the number of GO terms within each cluster, and the color indicates the log10-transformed adjusted *p*-value of the enrichment analysis.

**Figure 2 insects-16-00325-f002:**
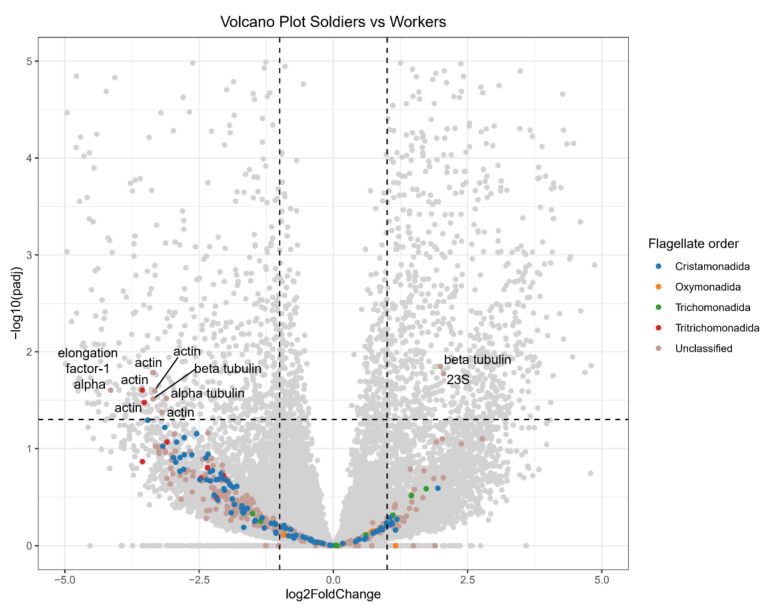
Volcano plot comparing flagellate transcript abundance between soldiers and workers. Grey points represent transcripts of putative termite or unclear sources. The black dashed lines indicate the criteria for differential abundance: fold change >2 or <0.5, adjusted *p*-value < 0.05. Transcripts on the right are soldier-biased, while those on the left are worker-biased.

**Figure 3 insects-16-00325-f003:**
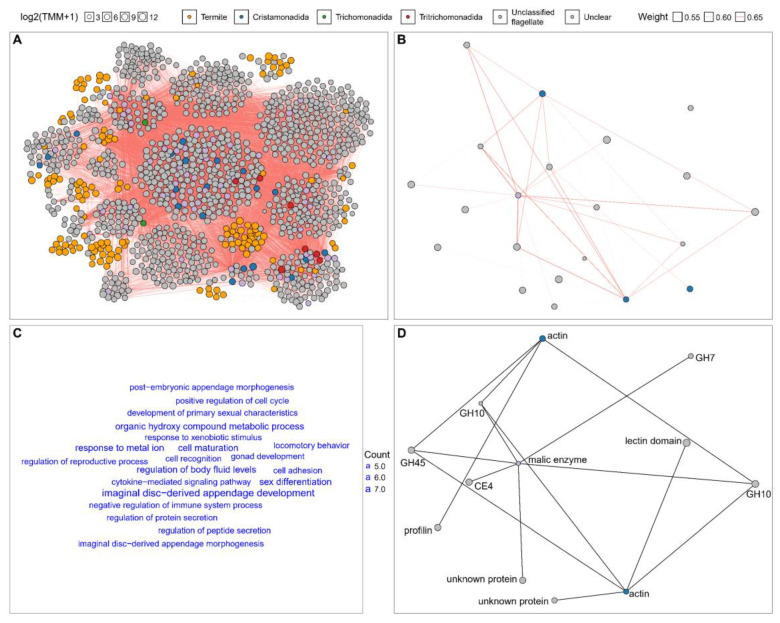
Co-expression patterns of host–flagellate transcripts in worker and soldier termites. The co-expression network shows strong correlations between host–flagellate transcripts in (**A**) workers and (**B**) soldiers. Only transcript pairs with a correlation (edge weight) ≥ 0.5 between putative flagellate and other transcripts are shown. Transcripts within the same co-expression modules are clustered together. (**C**) Gene Ontology (GO) terms in the Biological Process (BP) category for putative termite transcripts that are strongly correlated with putative flagellate transcripts. Only GO terms with more than 5 associated termite transcripts are shown. (**D**) Strong co-expression correlations between putative flagellate and other transcripts in both workers and soldiers. The network topology resembles that of the soldier network.

**Table 1 insects-16-00325-t001:** Worker-biased and soldier-biased biomarker transcripts detected using Least Absolute Shrinkage and Selection Operator (LASSO) regression. Positive LASSO regression coefficients indicate soldier-biased transcripts, while negative coefficients indicate worker-biased transcripts. Expression levels in worker and soldier samples were normalized using the Trimmed Mean of M-values (TMM). Differentially expressed transcripts (DETs) were identified using DESeq2.

Transcript	Putative Source	Coefficient	Average Worker TMM	Average Soldier TMM	DET
inositol 1,4,5-trisphosphate receptor mRNA	termite	−0.00015	97.58	18.41	no
resistance to inhibitors of cholinesterase protein 3 (RIC3) mRNA	termite	0.00121	803.25	2862.81	yes
T-box transcription factor TBX20 mRNA	termite	0.01148	24.28	127.13	yes
CAP-Gly domain-containing linker protein 2 (CLIP2) mRNA	termite	0.00022	187.64	1157.88	yes
periaxin mRNA	termite	−0.00178	172.28	3.17	yes
regulator of microtubule dynamics protein 1 (RMDN1) mRNA	termite	−0.00261	346.49	30.11	yes
CAP-Gly domain-containing linker protein 2 (CLIP2) mRNA	unclear	0.00105	656.38	3365.92	yes
choline transporter-like protein 2 (CTL2) mRNA	termite	−0.00071	960.42	141.31	yes
serine/threonine-protein kinase mRNA	unclear	4.56096	125.45	1099.15	no
Kazal-type serine protease inhibitor mRNA	unclear	1.95818	149.86	1717.32	yes
sorting nexin-29 (SNX29) mRNA	termite	−0.01008	173.60	12.24	yes
pancreatic triacylglycerol lipase (PTL) mRNA	termite	0.00021	287.63	2320.95	yes

**Table 2 insects-16-00325-t002:** Order and gene identification of flagellate transcripts. The data in this table represent the number of transcripts identified.

Gene	Cristamonadida	Oxymonadida	Trichomonadida	Tritrichomonadida	Unclassified Order	Total
18S rRNA	10	7	2	0	5	24
actin	63	0	4	11	76	154
alpha tubulin	0	1	0	1	48	50
beta tubulin	0	0	0	0	56	56
elongation factor-1 alpha (EF1α)	16	0	1	0	10	27
glyceraldehyde-3-phosphate dehydrogenase (GAPDH)	5	0	0	0	16	21
others	0	0	3	0	98	101
total	94	8	10	12	309	433

**Table 3 insects-16-00325-t003:** Genus-level identification of flagellate transcripts. The percent identity of the best hit was determined by BLASTN against the NT database. Transcript abundance in worker and soldier samples was normalized using the Trimmed Mean of M-values (TMM). The adjusted *p*-value for differential expression analysis was calculated using DESeq2.

Gene	Genus	Order	Percent Identity of Best Hit	Average Worker TMM	Average Soldier TMM	Adjusted *p*-Value
18S rRNA	*Devescovina*	Cristamonadida	92.55	1011.79	375.31	0.55
18S rRNA	*Devescovina*	Cristamonadida	95.75	105.92	64.66	0.78
18S rRNA	*Devescovina*	Cristamonadida	95.67	200.04	88.15	0.59
18S rRNA	*Devescovina*	Cristamonadida	91.80	197.95	175.36	0.95
18S rRNA	*Oxymonas*	Oxymonadida	94.78	547.68	908.43	0.71
18S rRNA	*Oxymonas*	Oxymonadida	92.41	90.92	178.40	0.59
18S rRNA	*Oxymonas*	Oxymonadida	96.83	147.28	271.46	0.70
18S rRNA	*Oxymonas*	Oxymonadida	97.46	869.33	1632.84	0.66
18S rRNA	*Oxymonas*	Oxymonadida	97.38	442.22	830.14	0.65
18S rRNA	*Blattamonas*	Oxymonadida	99.12	1468.43	2481.67	0.71

## Data Availability

Raw reads and assembly of the transcriptomes were deposited at NCBI SRA and TSA database, respectively (BioProject accession: PRJNA1224768).

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
