# Peer review of "Comparison Between Worker and Soldier Transcriptomes of Termite Neotermes binovatus Reveals Caste Specialization of Host–Flagellate Symbiotic System"

_insects, 2025, doi:10.3390/insects16030325_

Round 1

Reviewer 1 Report

Comments and Suggestions for Authors

Review of the manuscript entitled “Comparison Between Worker and Soldier Transcriptomes of a Drywood Termite Species Reveals Caste Specialization of the Host-Flagellate Symbiotic System” by Huang et al.

The authors performed short-read RNA-seq using RNA extracted from the whole bodies of workers and soldiers in Neotermes sp. They conducted differential expression analysis between workers and soldiers, as well as correlation analysis of transcript expression levels between intestinal protozoa and termites.

Differential expression analysis between castes using RNA-seq data has been performed in many termite species before, and this study does not provide many new discoveries but supports previous findings. On the other hand, the comparison of flagellate transcript levels between castes and the correlation analysis between termite and flagellate gene expression are relatively novel, providing new insights.

The bioinformatics analysis in this study uses orthodox tools and is scientifically sound. However, in other parts, there are points that require improvement, as shown below.

L. 44-46 Soldiers, which develop ...

Not all soldiers possess powerful mandibles. This sentence requires modification.

L. 95-96 Ten workers and ten soldiers…

The original colony of each individual should be mentioned. In the study of social insects, generally speaking, individuals from the same colony share the same background in terms of genetic composition, intestinal protists, and so on, and thus, individuals from a single colony are not statistically independent but spseudo-replication. If individuals used in this study originated from a single colony, the authors must mention about validity of it.

L. 96-97 Through morphological identification…

References for the morphological identification should be shown.

L. 102-104 Only RNA samples …

Numbers of worker and soldier RNA samples must be shown. In addition, the library preparation method for Illumina sequencing must be described.

L. 339-340 Flagellate transcript abundance …

The authors conclude that flagellate transcript abundance tends to be higher in workers, but this interpretation should be revised. As shown by the statistical analysis, there were not significant differences between flagellate transcript abundances of workers and those of soldiers, and more than half of the gene transcripts shown in Table 3 are more abundant in soldiers than in workers.

Comments on the Quality of English Language

As indicated in the review comments for lines 44-46, there are English expressions that could potentially lead to misunderstandings.

Author Response

Thank you for your appreciation of our research and helpful comments for improving our manuscript. Details about your comments and suggestions can be found below in our point-to-point responses. Please find our detailed point-by-point responses in the accompanying document, with the original comment in black and our responses in red font.

Reviewer 2 Report

Comments and Suggestions for Authors

Huang et al. provide a transcriptomic analysis of worker versus soldier castes in a Neotermes sp., highlighting caste-specific gene expression and host-flagellate interactions. The research is well-motivated, and the transcriptomic sequencing is done in high depth. However, experimental design and data analysis give rise to serious concerns about statistical robustness, biological interpretation, and methodological transparency. The results are overemphasized without sufficient reasons, and the manuscript contains significant linguistic flaws, which makes it hard to read. General scientific rating: 65/100. Language quality: 6/10.

Major points

Sample size is not well justified, with only ten members per caste selected for transcriptome analysis. Power analysis is not provided, and hence the study can or cannot be sufficiently powered to detect significant differential expression.

Transcriptome assembly generates a very high number of contigs (1,042,913), which is high and indicates fragmentation and/or poor quality of the assembly. Transcript redundancy filtering is not mentioned at all, which can lead to overestimation of differentially expressed gene numbers.

Flagellate transcript classification relies entirely on BLAST-based taxonomic assignment, which can be wrong with a narrow database of references for termite symbionts. Phylogenetic confirmation of transcript origin is not performed.

Statistical methods of differential expression analysis are not precise about normalization procedures and variance stabilization. When DESeq2 is used, batch effects or latent confounders that may impact results are not accounted for.

The manuscript reports tight co-expression of termite and flagellate genes but fails to validate these observations with functional assays. Correlation does not imply causation, and the biological significance of these interactions is speculative.

The assertion overinterprets caste-specific gene expression with no allowance for the possibility of confounding effects of developmental stage, feeding status, or individual variation that are not accounted for in the research.

Key conclusions, such as the proposed dependence of soldiers on worker-processed food, are inferred from transcriptomic trends rather than direct measurement of metabolism or physiology.

Minor points

Figures are not fully informative without important statistical annotations, which render it difficult to make an estimate of the significance and reliability of reported differences.

The manuscript breaks convention in journal style, particularly reference citations and figure legends.

There are some unclear sentences and some grammatical errors that reduce clarity, e.g., "This suggests a possibly comparable phenotype" instead of a categorical statement from data.

Gene ontology enrichment analyses are presented without sufficient explanation of how they assist in demonstrating functional divergence between castes.

Neotermes sp. identification is based on morphology and COX2 data only but without whole-genome validation, so species designation is provisional.

The protocols lack procedures for RNA integrity control other than standard Nanodrop measurements, which can be too low for high-throughput transcriptomics.

The manuscript has excellent transcriptomic significance for Neotermes sp. caste specialization, but conclusions need to be experimentally validated. Future work will require tighter controls on developmental and physiological parameters, phylogenetic confirmation for symbiont transcripts, and functional assays for validation of presumed interactions with the host and flagellate.

The work has a valuable dataset but fails to meet criteria for proper transcriptomic analysis and biological inference warranted. It should not be publishable in the current state but with significant improvements in experimental design, statistical consideration, and functional validation.

Comments on the Quality of English Language

Several ambiguous sentences and grammatical errors reduce clarity.

Author Response

(The authors gave the same response as above.)

Round 2

Reviewer 2 Report

Comments and Suggestions for Authors

The authors have responded to my comments effectively, providing convincing arguments. They rationalized the sample size based on comparable studies, explained the quality of the transcriptome assembly, and improved statistical methodologies by detailing the normalization procedures. The correlation vs. causation issue was clearly acknowledged, with a clear statement emphasizing the necessity of experimental verification. They clarified figure legends and reworded confusing terminology for easier reading. Although certain restrictions still apply, e.g., the lack of phylogenetic confirmation and some experimental controls, they were pointed out, and future research was suggested to rectify them. In general, the authors' attempts adequately address my concerns, and the manuscript is acceptable for publication with minor improvements.

One point is remaining, although BLAST-based taxonomic assignment is a quick way to classify, it does not determine evolutionary relationships and its accuracy is relative to the completeness of the database and may therefore result in misclassifications. For more accurate taxonomic resolution, phylogenetic methods like Maximum Likelihood or Bayesian inference need to be employed in future analyses to verify transcript origins.
The application of more than one classification tool, for example, Kraken2 or PhyloPythia, will minimize the opportunity for misclassification and enhance precision.

BLAST analysis is limited by reference database sizes. Without representative termite symbiont genomes, origins of transcripts may be incorrectly identified. This limitation is employed to further highlight the necessity for additional sequencing effort on microbe species that are symbiotic with termites.

HOWEVER, I'm satisficed by the overall result! Well done!